# Brief Communication: Modified KdV equation for Rossby-Khantadze waves in a sheared zonal flow of the ionospheric E-layer

**Laila Zafar Kahlon[1]\*, Hassan Amir Shah[1], Tamaz David Kaladze[2, 3], Qura Tul Ain[1], Syed Assad Ul Azeem Bukhari[1],**

[1]Physics Department, Forman Christian College (a Chartered University), Lahore 54600, Pakistan
[2]I. Vekua Institute of Applied Mathematics, Tbilisi State University, 2 University str, Tbilisi 0186, Georgia
[3]E. Andronikashvili Institute of Physics, I. Javakhishvili Tbilisi State University, Tbilisi 0128, Georgia

\*Corresponding author: Email address: lailakahlon@fccollege.edu.pk (Laila Zafar Kahlon)

## Abstract

The system of nonlinear equations for electromagnetic Rossby-Khantadze waves in a weakly ionized conductive ionospheric E-layer plasma with sheared zonal flow is given. Use of multiple-scale analysis allows reduction of obtained set of equations to (1+1)D nonlinear modified KdV (mKdV) equation with cubic nonlinearity describing the propagation of solitary Rossby-Khantadze solitons.

**Keywords:** Rossby-Khantadze waves; nonlinearity, sheared zonal flow

## 1. Introduction

Different satellite and ground-based investigations indicate presence of zonal flows in various atmospheric regions around the Earth (Pedlosky, 1987). The reason for the existence of zonal flows is the non-uniform warming of the Earth's atmospheric regions by the sun. The presence of sheared flow along the meridians with inhomogeneous velocity, is closely connected with the ultra-low-frequency perturbations in ionospheric E and F regions of the ionosphere (Satoh, 2004; Shukla et al., 2003; Onishchenko et al. 2004; Kaladze et al., 2007; Kaladze et al., 2008). Effects of sheared flow appear in linear and nonlinear properties of the waves, and conditions suitable for that are available in Earth's ionosphere. This gives rise to a variety of nonlinear phenomena like formation of solitary structures (solitons, vortices, zonal flows, etc.).

Due to a significant role in the global atmospheric circulation Rossby waves attract special scientific attention in connection with sheared zonal flows. Note that spatial nonhomogeneity of Coriolis parameter alongwith ambient geomagnetic field along the meridians causes the propagation of such coupled Rossby-Khantadze (RK) electromagnetic (EM) waves (see e.g. Kaladze et al. 2011). The generation of sheared RK EM planetary vortices in the ionospheric E-region also discussed (Kaladze et al., 2011; Kaladze et al., 2014). It was revealed, that propagation of coupled EM RK waves could be self-organized into solitary dipolar vortices and the possibility of the generation of intensive magnetic field is shown. In recent decades, several nonlinear phenomena related to the excitation of sheared zonal flows by EM Rossby waves were investigated. Taking into account Reynolds stresses zonal flow generations by short wavelength EM Rossby waves studied (Shukla et al., 2003; Onishchenko et al. 2004). The zonal flow' generation in the ionospheric E-layer by Rossby waves revealed

by Kaladze et al. (2007). Such nonlinear Rossby wave structures broken into numerous parts depends on the zonal flow energy (Kaladze et al., 2008). Numerical work on EM RK waves with sheared zonal flow in ionospheric E-plasma was found as well (Futatani et al., 2013, 2015). In this work it was pointed out the splitting of vortices, where the energy is transported by sheared flow into multiple pieces. Equatorial Rossby wave solitons under the action of sheared flows were also discussed (Qiang et al., 2001) and the existence of solitons was confirmed by the observations of *Freja and Viking satellites* (Qiang et al., 2001; Bostrom, 1992; Dovner et al. 1994; Lindqvist et al., 1994). Jian et al., (2009) investigated nonlinear propagation of Rossby waves in stratified neutral fluids with zonal shear flow and obtained modified Korteweg-de Vries (mKdV) equation with cubic nonlinearity. Generation of the zonal flow alongwith magnetic field in the ionospheric E-plasma by Rossby-Khantadze EM planetary waves also discussed (Kaladze et al. 2012, Kahlon and Kaladze 2015). Possibility of magnetic field generation of $10^3$ nT is predicted. Kaladze et al. (2019) investigated nonlinear interaction of magnetized electrostatic Rossby waves with sheared zonal flows in the Earth's ionospheric E-layer and developed the modified Korteweg-de Vries (mKdV) equation having cubic nonlinearity describing propagation of appropriate solitons. Some premises of the possibility of existence of planetary Rossby waves in the dynamo E-area of weakly ionized ionosphere and corresponding experimental interpretation was discussed by Forbes, 1996. Also, Vukcevic M. and Popovic L. Č., (2020) pointed out the possibility of many soliton structure formations at different latitudes, and at diverse ionospheric layers. Direct observations of such soliton structures from the surface of Earth or onboard the satellites are discussed.

In the given manuscript, we generalize mentioned above results for the weakly ionized conducting ionospheric E-region plasma by incorporating along with stream-function evolution of geomagnetic field for electromagnetic RK waves, which to the best of our knowledge was not reported so far and thus provides novelty to this work. In Sec. 2, from the obtained system of nonlinear two-dimensional equations by using the multiple scale analysis and perturbation approach we derive one-dimensional mKdV equation with cubic nonlinearity describing solitary Rossby-Khantadze waves dynamics along with zonal (shear) flows. Sec. 3 includes the discussion of the results.

## 2. Mathematical Preliminaries

We consider partially ionized E-ionospheric region consisting of small concentration of electrons, ions and bulk of neutral particles, where such ionospheric plasma is enclosed in a spatially inhomogeneous geomagnetic field $\boldsymbol{B}_0 = (0, B_{0y}, B_{0z})$ and the Earth's angular velocity $\boldsymbol{\Omega} = (0, \Omega_{0y}, \Omega_{0z})$. In weakly ionized ionospheric E-layer plasma, we consider two-dimensional' wave motion $\mathbf{v} = (u, \mathrm{v}, 0)$, where $u = -\frac{\partial \psi}{\partial y}$, $\mathrm{v} = \frac{\partial \psi}{\partial x}$, and $\psi(x, y, t)$ is the stream function.

We consider a local Cartesian system of coordinates with zonal x, latitudinal y, and z in local vertical direction. Then the nonlinear behavior of the sheared electromagnetic Rossby-Khantadze waves can be narrated by the following 2D system of equations (e.g. Kaladze et al., 2014),

$$\begin{cases} \frac{\partial \Delta \psi}{\partial t} + \beta \frac{\partial \psi}{\partial x} + \mathrm{J}(\psi, \Delta \psi) - \frac{1}{\mu_0 \rho} \beta_B \frac{\partial h}{\partial x} = 0 \,, \\ \frac{\partial h}{\partial t} + \mathrm{J}(\psi, h) + \beta_B \frac{\partial \psi}{\partial x} + c_B \frac{\partial h}{\partial x} = 0 \,, \end{cases} \quad (1)$$

96        The first equation describes the evolution of the z-component of vorticity ($\zeta_z = \boldsymbol{e}_z \cdot$
$\nabla \times \boldsymbol{v} = \Delta\psi$) of the singly fluid momentum equation under the action of the geomagnetic field,
$\boldsymbol{v}$ is the velocity of the incompressible neutral gas. The second equation is the z-component of
the perturbed magnetic induction h obtained through Faraday's law, and $\beta = \frac{\partial f}{\partial y} = \frac{2\partial \Omega_{0z}}{\partial y}$
describes the latitudinal inhomogeneity of angular velocity. Also the parameter $c_B = \beta_B / e n \mu_0$
with $\beta_B = \frac{\partial B_{0z}}{\partial y}$, describes the latitudinal inhomogeneity in the background magnetic field, $n$
is the number density of the charged particles, $\mu_0$ is the magnetic permeability and $J(a, b) =$
$\frac{\partial a}{\partial x} \frac{\partial b}{\partial y} - \frac{\partial a}{\partial y} \frac{\partial b}{\partial x}$ is the Jacobian (responsible for the vector nonlinearity) and $\Delta = \partial_x^2 + \partial_y^2$. Note
that the small concentration of charged particles (compared to the neutral particles) gives the
contribution only in the inductive current (Kaladze, et al. 2013a, 2013b). It should also be noted
that the ambient magnetic field and Coriolis parameter are spatially inhomogeneous, (Kaladze,
et al., 2014). Details on the system (1) can be found in Kaladze, et al. (2012).
109       The boundary conditions that are fulfilled for this system are given as,
$$\psi(0) = \psi(1) = 0, \tag{2}$$
which represents the flow's edges, specifically along the south and north direction (Pedlosky
(1987); Satoh (2004)).

**2.1 Perturbation and weakly nonlinear approach**
118       The background stream function is considered in the following manner:
$$\Psi(y) = -\int [U(y) - c_0] dy. \tag{3}$$
Here $U(y)$ describes the basic background flow with $c_0$ as a constant eigenvalue. The whole
stream function $\psi$ is considered as the sum of background (zonal flow) stream function $\Psi(y)$
and a disturbed stream $\psi'$ function. This assumption makes it a weakly nonlinear system, that
is the subject of this study. While the perturbed magnetic field is also characterized by a small
a parameter $\varepsilon$. Therefore the stream function and the magnetic perturbations takes the form,
$$\psi = \Psi(y) + \varepsilon\psi' = -\int [U(y) - c_0] dy + \varepsilon\psi',$$
$$h = \varepsilon h' \tag{4}$$
where $\varepsilon \ll 1$ is a small parameter indicating that the perturbed quantities are small compared
to the background parameters.
130       Using Eq (4) into (1) gives

$$\begin{cases} \frac{\partial \Delta\psi'}{\partial t} + (U(y) - c_0)\frac{\partial \Delta\psi'}{\partial x} + (\beta - U'')\frac{\partial \psi'}{\partial x} + \frac{\beta_B}{\mu_0 \rho}\frac{\partial h'}{\partial x} + \varepsilon J(\psi', \Delta\psi') = 0, \\ \frac{\partial h'}{\partial t} + \varepsilon J(\psi', h') + (U(y) - c_0)\frac{\partial h'}{\partial x} + \beta_B\frac{\partial \psi'}{\partial x} + c_B\frac{\partial h'}{\partial x} = 0. \end{cases} \tag{5}$$
where $U'' = \frac{d^2 U}{dy^2}$.

By using the multiple scale analysis, we obtain the asymptotic solution where we take the spatial and temporal parameters as $X = \varepsilon x$ and time $T = \varepsilon^3 t$ respectively. Further by eliminating $h'$ from 5(b) into 5(a) we get the single equation for $\psi'$

$$\mathcal{L}_0(\psi) + \varepsilon^2 \mathcal{L}_1(\psi) + \varepsilon\, J\left(\psi, \frac{\partial^2 \psi}{\partial y^2}\right) + \varepsilon^3\, J\left(\psi, \frac{\partial^2 \psi}{\partial X^2}\right) + \varepsilon^4\, \frac{\partial^3 \psi}{\partial T \partial X^2} = 0 . \qquad (6)$$

In Eq. (6) the prime on the perturbed stream function is dropped, and the following linear differential operators are introduced

$$\mathcal{L}_0 = \left[(U - c_0)\frac{\partial^2}{\partial y^2} + p(y) + \frac{\alpha(y)}{U - c_0 + c_B}\right]\frac{\partial}{\partial x}, \quad \mathcal{L}_1 = \frac{\partial}{\partial T}\frac{\partial^2}{\partial y^2} + (U - c_0)\frac{\partial^3}{\partial X^3} , \qquad (7)$$

where $\alpha(y) = \frac{\beta_B^2}{\mu_0\,\rho}$ and $p(y) = \beta - U''$. Here the parameter $\alpha$ takes into account the spatial inhomogenity of the background magnetic field which was not considered before in Kaladze et al. (2019).

Furthermore, we expand the stream function $\psi$ (in series with respect to the $\varepsilon$) as:

$$\psi = \psi_0 + \varepsilon\, \psi_1 + \varepsilon^2 \psi_2 + \cdots . \qquad (8)$$

By using Eq. (8) into Eq. (6), we obtain from the lowest order $O(\varepsilon^0)$, the following equations,

$$\mathcal{L}_0[\psi_0] = 0, \quad \text{with} \quad \psi_0 = 0 \ for \ y = 0,1 . \qquad (9)$$

The above equation (9) is a linear differential equation. By performing a separation of variables method for $\psi_0 = A(X, T)\,\Phi_0(y)$ into this form and substitute it into Eq. (7) we get the following equation with conditions of boundary:

$$\left(\frac{d^2}{dy^2} + \frac{p(y)}{U - c_0} + \frac{\alpha(y)}{(U - c_0)(U - c_0 + c_B)}\right)\Phi_0 = 0 , \quad \text{with} \quad \Phi_0(0) = \Phi_0(1) = 0. \qquad (10)$$

Here we consider $U - c_0 \neq 0 \ and \ U - c_0 + c_B \neq 0$. This is an eigenvalue problem for eigen value $c_0$. By specifying $p(y)$ and $\alpha(y)$, $\Phi_0(y)$ can be found. Since $p(y)$ and $\alpha(y)$ have dependence on the variable y, it is not easy to solve this eigen value problem analytically. From the lowest order $O(\varepsilon^0)$, we see that the problem is time independent, but cannot be analytically solved as we have not substituted any definite dependence on y for the parameters $p(y)$ and $\alpha(y)$. Thus, in order to get more details about the amplitude of these waves, we go to the next order i.e. $O(\varepsilon^1)$ from Eqs. (7) and (8), we obtain

$$\mathcal{L}_0[\psi_1] = -J\left(\psi_0, \frac{\partial^2 \psi_0}{\partial y^2}\right) \equiv F_1 = A\frac{\partial A}{\partial X}\left(\frac{p(y)}{U - c_0} + \frac{\alpha}{(U - c_0)(U - c_0 + c_B)}\right)_y \Phi_0^2 , \qquad (11)$$

Furthermore, we carry out a separation of variables in the following manner $\psi_1 = \frac{1}{2} A^2(X, T)\,\Phi_1(y)$ for non-singular neutral solutions into (11)

$$\left(\frac{d^2}{dy^2} + \frac{p(y)}{U - c_0} + \frac{\alpha}{(U - c_0)(U - c_0 + c_B)}\right)\Phi_1 = \left(\frac{p(y)}{U - c_0} + \frac{\alpha}{(U - c_0)(U - c_0 + c_B)}\right)_y \frac{\Phi_0^2}{(U - c_0)}, \qquad (12)$$

For the given boundary conditions $\Phi_1(0) = \Phi_1(1) = 0$. To get amplitude we solve Eqs. (7)
and (8) in the next order i.e. $O(\varepsilon^2)$ which gives

$$\mathcal{L}_0[\psi_2] = -\mathcal{L}_1[\psi_0] - J\left(\psi_0, \frac{\partial^2 \psi_1}{\partial y^2}\right) - J\left(\psi_1, \frac{\partial^2 \psi_0}{\partial y^2}\right) \equiv F_2, \tag{13}$$

with $\psi_2(0) = \psi_2(1) = 0$.
Here it is pointed out that the dispersion effect, given in the definition of $\mathcal{L}_1$ competes with
weakly nonlinear effect, which appears through the Jacobian in Eq. (9).
Furthermore, we again perform a separation of variables, $\psi_2 = B(X,T)\Phi_2(y)$ and multiply
Eq. (13) by $\psi_0$ and integrate over $y$, which yields

$$\int_0^1 dy \; \frac{F_2}{U - c_0} \Phi_0 = 0 \;. \tag{14}$$

By substituting $F_2$ and using $\psi_1 = \frac{1}{2} A^2(X,T) \Phi_1(y)$ into Eq. (14) we get the modified KdV
(mKdV) equation (Kaladze et. al (2019))

$$\frac{\partial A}{\partial T} + N A^2 \frac{\partial A}{\partial X} + D \frac{\partial^3 A}{\partial X^3} = 0 \;. \tag{15}$$

This equation has a cubic nonlinearity, whereas the standard KdV equation has a quadratic
nonlinearity.
In Eq.(15) above

$$N = \frac{I_2}{I_0} \;, \qquad D = -\frac{I_1}{I_0} \;, \tag{16}$$

where

$$
\begin{cases}
I_0 = \int_0^1 dy \; \Phi_0^2(y) \left[ \frac{p(y)}{(U(y)-c_0)^2} + \frac{\alpha}{(U(y)-c_0)^2(U(y)-c_0+c_B)} \right], \\[2mm]
I_1 = \int_0^1 dy \; \Phi_0^2(y) \\[2mm]
I_2 = \int_0^1 dy \; \frac{\Phi_0^2(y)}{U(y)-c_0} \begin{cases} \frac{3}{2}\left(\frac{p(y)}{U(y)-c_0} + \frac{\alpha}{(U(y)-c_0)(U(y)-c_0+c_B)}\right)_y \Phi_1(y) \\[2mm] -\frac{1}{2}\Phi_0^2(y)\left[\left(\frac{p(y)}{U(y)-c_0} + \frac{\alpha}{(U(y)-c_0)(U(y)-c_0+c_B)}\right)_y \frac{1}{U(y)-c_0}\right]_y \end{cases}
\end{cases} \tag{17}
$$

207       Kaladze et al. (2019) and Jian et al. (2009) also obtained the same mKdV equation (15)
with cubic nonlinearity for Rossby waves and pointed out that the background flow shear is a
necessary condition for the existence of solitary waves, whereas in this work, we get the mKdV
for the Rossby-Khantadze waves where the coefficients have been modified by inclusion of
inhomogeneity in geomagnetic field. Moreover, the effect of shear basic flow on the spatial
structure, propagation velocity and wave width of solitary Rossby waves have been studied.
We would like to point out here the meridional dependence of functions $\beta(y)$, $\alpha(y)$ and $U(y)$,
that appears in the coefficients $N$ and $D$.
216       Amid numerous exact solutions of mKdV equation (15) (see e.g. Wazwaz (2009)), we
are interested in a soliton like traveling wave solution. The one-soliton solution of equation
(15) is

$$A(X,T) = \pm \sqrt{\frac{6c}{N}} \, sech\left(\sqrt{\frac{c}{D}}(X - cT)\right), \qquad (18)$$

where c is the traveling wave velocity, and the coefficients $N$ and $D$ are defined by Eqs. (16)-(17). In order for a wave to have an exact solitary solution associated to it, one needs to a robust equation like the KdV. Modified KdV, as well, has infinite conservation laws associated to them, and hence is integrable and contain one and N-soliton solution. Shown in the above equation is the one soliton solution of the mKdV. One can use the Hirota's method, where by using a suitable transformation, one converts the nonlinear equation into a bilinear equation, and then by using the Hirota's differential operator and solving the subsequent equation, one can obtain a multi-soliton solution. Some types of mKdV spatially periodic solutions (cnoidal solutions) discussed (Kevrekidis et al. 2004). It was noted that mKdV equation having nonlinear term may have an alternate sign. Properties of such difference also discussed.

### 3. Discussion

In the present paper, we have studied the nonlinear dynamics of large-scale electromagnetic Rossby-Khantadze waves with zonal flows in E-ionospheric plasma. Both the latitudinal inhomogeneities in angular velocity of the earth's rotation and the geomagnetic field are taken into account. The latitudinal inhomogeneity of the magnetic field is responsible for coupled Rossby–Khantadze waves. Such coupling results in an appearance of dispersion of Khantadze waves. To derive the nonlinear modified KdV we used the multiple scale analysis technique. From the lowest order of O ($\varepsilon^0$), we get an eigen-value problem with constant eigen-value $c_0$ along with the boundary conditions. The parameters $p(y)$ and $\alpha(y)$ have dependence on the variable y, making it not possible to solve this eigen value problem analytically. From the next order O ($\varepsilon^1$), by using separation of variables techniques and after doing some mathematical manipulations we arrive at the mKdV equation (15) with cubic nonlinearity of (1+1) dimension. Traveling wave solitary solution of this equation is given by Eq. (18), where the parameter $\sqrt{\frac{6c}{N}}$ describes the amplitude of solitary RK structures. The obtained coefficients $N$ and $D$ depend on the spatially inhomogeneous Coriolis force $\alpha(y)$ and background magnetic field $\beta(y)$, respectively.

In anticipation of future for the experimental observations of RK vortical motions in the weakly ionized ionospheric E-layer we expect the following characteristics. Apart from the ordinary Rossby waves electromagnetic RK perturbations generated by the latitudinal gradient of the geomagnetic field and represent the variation of the vortical electric field $\boldsymbol{E}_v = \mathbf{v}_D \times \boldsymbol{B}_0$, where $\mathbf{v}_D = \boldsymbol{E} \times \boldsymbol{B}_0 / B_0^2$ is the electron drift velocity. RK waves propagate along the latitude with the velocity $|c_B| \approx 2 - 20 \, km/s$. Frequency ($\omega = k_x c_B$) and the phase velocity $c_B$ depend on the number density of the charged particles and vary by one order of magnitude during the daytime and nighttime conditions (which is so suitable for experimental observations). Such perturbations have relatively high frequency $(10^4 - 10^{-1}) \, s^{-1}$ and have wavelengths $\sim 10^3 \, km$. Compared with the ordinary Rossby waves electromagnetic RK waves accompanied by the strong pulsations of the geomagnetic field 20-80 nT. Note that Khantadze waves in the middle and moderate latitudes observed at the launching of spacecrafts Burmaka, et al. (2006) and by the world network of ionospheric and magnetic observations Sharadze, et al. (1988); Sharadze, et al. (1989); Sharadze, (1991); Alperovich, et al. (2007). Forbes (1996) provides data analyses for discussing the penetration of Rossby type planetary waves effects into ionospheric dynamo E-region (100-170 km) and the electrodynamic interactions which ensue there.

RK waves are mainly of zonal type and observed mainly during magnetic storms alongwith sub-storms, artificial explosions, earthquakes, etc. They give valuable information on large-scale synoptic processes and about external sources as well as dynamical processes in the ionosphere. Therefore, theoretical investigations of electromagnetic Rossby type oscillations will provide valuable information for further ionospheric experimental investigations.

**AUTHOR DECLARATIONS:**

**Conflict of Interest**
The authors have no conflicts to disclose.

**Data Availability**
The data that support the findings of this study are available within the article.

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
