# Peer review of "Brief Communication: Modified KdV equation for Rossby-Khantadze waves in a sheared zonal flow of the ionospheric E-layer"

_Nonlinear Processes in Geophysics, 2023_

## Referee Comment (RC1)

**The report on the paper**
**"Modified KdV equation for Rossby-Khantadze waves in a sheared zonal flow of the ionospheric E-layers"**

The paper addresses the problem of propagation of nonlinear Rossby waves in the presence of zonal flow and magnetic field. The authors used the reductive perturbation method to derive the mKdV equation describing these waves. Previously this equation has been already derived, however in the absence of magnetic field. Hence, potentially, the paper can be published. However it needs a very substantial improvement.

**Minor comments**

1. In Abstract the authors write "in a weakly ionospheric plasma". I guess it should be "in a weakly conducting ionospheric plasma".

2. In Sect. 2, define $n$ and $\mu_0$.

3. The sentence after Eq. (5) does not make very much sense.

4. The right term is not "apostrophe" but "prime".

My main concern is English. Sometimes it is so poor that it is even difficult to understand what the authors wanted to say. I suggest that the authors look for help of somebody with good command in English.

Summarising, I thin that the paper needs a serious revision.

---

## Author Comment (AC1)

Response to the Reviewer 1:

Dear Referee,

We are very thankful to you for your valuable suggestions. By keeping your comments in mind, we improve our manuscript.

**Minor changes:**
1. **In Abstract the authors write "in a weakly ionospheric plasma". I guess it should be "in a weakly conducting ionospheric plasma".**
Reply: In p.1, lines 18-22, *we improve the line "in a weakly ionospheric plasma" with "in a weakly conducting ionospheric plasma".*
2. **In Sect. 2, define n and μ0.**
Reply: in p.2, line 99, the symbols n and μ0 have been defined.
3. **The sentence after Eq. (5) does not make very much sense.**
Reply: The confusing sentence "It is also noted that parameter ε involves magnitude of nonlinear products" has been removed.
4. **The right term is not "apostrophe" but "prime".**
Reply: The word "apostrophe" has been replaced with "prime".

**Major revision:**

5. *In p.1., lines 18-22, the abstract has been improved.*
6. *In p.1., line 27, the word "indicates" has been replaced with "indicate".*
7. *In p.1., line 29, the word "the" has been added after "various atmospheric …".*
8. *In p.1., lines 27-36, The following sentences "In E and F regions of ionosphere ..." have been rephrased with "The presence of sheared…".*
9. *In p.1., lines 38-44, the following sentences "In the past decade, several nonlinear phenomena were …" have been rephrased with "In recent decades, several nonlinear phenomena related …".*
10. *In p.1, lines 46-50, the following lines "the effects of the zonal (sheared) flows on Rossby nonlinear structures" have been rephrased with "The authors considered the effects of the zonal flows on nonlinear structures in Rossby waves and …".*
11. *In p.2., lines 51-54, the following sentence "Recently, it is implied that such coupled Rossby-Khantadze …" has been rephrased with "More recently, it was seen, that propagation of coupled Rossby-Khantadze (RK) waves…".*
12. *In p.2., lines 51-54, the following sentence "Recently, it is implied that such coupled Rossby-Khantadze …" has been rephrased with "More recently, it was seen, that propagation of coupled Rossby-Khantadze (RK) waves…".*
13. *In p.2., line 58, the word "were" is replaced by "was" and the word "as well" added after the word "investigated".*
14. *In p.2., lines 54-56, the following sentence "In the present work, the spatially inhomogeneous …" has been rephrased with "The spatially inhomogeneous Coriolis parameter and ambient magnetic field along the meridians …".*
15. *In p.2., lines 58-60, the following sentence "In this work the splitting of the vortices which …" has been rephrased with "In his work he has pointed out the splitting of …".*
16. *In p.2., lines 65-67, the following sentence "Kaladze et al. (2009) studied the solitary properties ..." has been rephrased with "Earlier, Kaladze et al. (2009) had earlier investigated the properties of ...".*

**17.** *In p. 2., lines 67-68, the following words "studied (Jian et al., 2009)." have been replaced with "done by Jian et al., (2009)."*

**18.** *In p. 2., lines 69-71, the following lines "The present problem is not reported before and the novelty ..." have been replaced with "In the present paper, we have considered the effect of magnetic field ...."*

**19.** *In p.2., lines 72, the words "partially ionized ionospheric E-region plasma." has been changed with "partially ionized conducting plasma, found in the ionospheric E-region".*

**20.** *In p.2., lines 74-77, the following sentence "In Sec. 2, by using multiple scale ..." has been rephrased with "In Sec. 2, by using the multiple scale analysis and perturbation approach from a system of …".*

**21.** *In p.2., lines 82-87, the following paragraphs "We have considered weakly conductive E-ionospheric …" is replaced with "We consider weakly ionized E-ionospheric region …"*

**22.** *In p.2., lines 88-93, the following paragraphs "The nonlinear behavior of considered sheared Rossby-Khantadze waves is pointed out …" is replaced with "The nonlinear behavior of the sheared Rossby-Khantadze waves …".*

**23.** *In p.3., lines 97-100, the following sentence "where h represents the z-component …" is rephrased with "In Eq. (1), $h$ represents the z-component …".*

**24.** *In p.3., line 116, the word "considered as" has been added after "is".*

**25.** *In p.3., line 117-118, the word ".... forms a weak nonlinear problem, ..." is" is replaced with "Which forms a weakly nonlinear system, …".*

**26.** *In p. 4, lines 166-168, the following lines "... ,we have used a couple of approximations the first ..." has been replaced with "... , we see that the problem is time independent, but cannot be analytically solved ..."*

**27.** *In p. 5, line 220, the word "responsible" is corrected with "responsible".*

---

## Author Comment (AC2)

Dear Editor/Reviewer,

The derivation of the system (1) is not the goal of our manuscript. It is discussed in many earlier published papers. To this end, we added most preferable reference (T.D. Kaladze, G.D. Aburjania, O.A. Kharshiladze, W. Horton, Y.-H. Kim, Theory of magnetized Rossby waves in the ionospheric E layer, J. Geophys. Res., v. **109**, A05302, doi: 10.1029/2003JA010049, 2004), where the reader can find answer about the system (1).

Meanwhile in the page #2, line 91, paragraph before the system (1) we have added two sentences in this connection: Here we obtained the following system of Eqs. (1) under the assumption that electron and ion flows due to the small concentration number (compared to the neutral particles) gives the contribution only in the inductive current (Kaladze, et al. 2013). The quantity $\zeta_z = \boldsymbol{e}_z \cdot \nabla \times \mathbf{v}$ is the z-component of the vorticity.

In page#7, line 286, the following reference has been added "[19] Kaladze T.D., Horton W., Kahlon L.Z., Pokhotelov O., and Onishchenko O., Zonal flows and magnetic fields driven by large-amplitude Rossby-Alfvén-Khantadze waves in the E-layer ionosphere**,** J. Geophys. Res.: Space Physics **118**, 1-12 2013."

P.S. The BRIEF COMMUNICATION status of the manuscript gives no possibility to discuss the questions set by reviewer in details.

Also, if our reviewer is familiar with any other new ionospheric equations, we are requesting to provide us the reference. As to the nonlinearity it is presented by Jacobian vector nonlinearity, which always exists as convective derivative of the z-component of the vorticity.

---

## Referee Report (RR1)

Review of the Revised Manuscript:

1. Language and Presentation:

The revised manuscript has shown significant improvements in terms of language and clarity. The content is structured coherently, and the language is refined. The previously identified issues, where well-articulated sections were interspersed with less clear segments, have been effectively addressed. The manuscript now maintains a consistent and polished writing style, enhancing its overall readability.

2. Contextual Framework and Motivation:

The manuscript has made commendable strides in establishing its context and motivation. By referencing seminal works, it provides a robust backdrop for the study, emphasizing the importance of electromagnetic Rossby-Khantadze waves in ionospheric research. This revision has successfully addressed prior concerns, offering a richer background and highlighting the research's significance.

3. Mathematical Exploration:

The manuscript delves deeply into the mathematical aspects, particularly focusing on the modified KdV (mKdV) equation and its solitary wave solution. The discussion around the derived MKdV equation and its association with quadratic nonlinearity is noteworthy. However, the manuscript could benefit from a more detailed exploration of the vanishing quadratic nonlinearity and the potential interplay of both quadratic and cubic nonlinearities in a perturbed system. A clear, step-by-step justification for these mathematical nuances would elevate the manuscript's quality.

4. Emphasis on mKdV Equation:

While the manuscript acknowledges the mKdV equation, citing works like Kaladze et al. (2009) and Jian et al. (2009), it falls short in emphasizing its nature as an exactly solvable equation with diverse solution methods. The authors' inclination towards the soliton-like solution is evident, but a clear rationale for this choice over other potential solutions is missing.

5. Significance of Discussed Waves:

The manuscript underscores the importance of the discussed waves, particularly highlighting their zonal type and relevance during specific atmospheric events. These waves provide valuable insights into the ionosphere's large-scale processes and dynamics. However, the manuscript could delve deeper into the scales of these solitary waves and their overarching significance.

Recommendations:

To further enhance the manuscript, it's crucial to provide a comprehensive exploration of the mathematical nuances, especially around the mKdV equation, and to delve deeper into the significance of the discussed waves. Addressing these aspects will ensure a holistic understanding and elevate the manuscript's academic standing.

---

## Referee Report (RR2)

**Referee report on Brief Communication: «Modified KdV equation for Rossby-Khantadze waves in a sheared zonal flow of the ionospheric E-layer».**

**Authors made important revision in manuscript. I recommend manuscript for publication.**

---

## Author Response (AR2)

Dear Editor,

We have made major changes in our manuscript according to your comments:

1) In line 18, we added "electromagnetic" before Rossby-Khantadze;
2) In line 20, we removed "so called";
3) In line 21 we changed the last word "waves" by "solitons";
4) In line 19 at the end of the 1st sentence we added "zonal flow in the ionospheric E-layer is given";

So, the revised abstract is as follows:

**Abstract**

The system of nonlinear equations for electromagnetic Rossby-Khantadze waves in a weakly ionized conductive ionospheric plasma with sheared zonal flow in the ionospheric E-layer is given. Use of multiple-scale analysis allows reduction of obtained set of equations to (1+1)D nonlinear modified KdV (MKdV) equation describing the propagation of solitary Rossby-Khantadze solitons.

5) In line 59 we changed "he has" by "it was pointed";
6) We continued line 44 by line 45;
7) In line 67 instead the abbreviation we added "modified Korteweg-de Vries (mKdV) equation";
8) In lines 69, 74 we changed MKdV by mKdV;
9) In Line 51, we changed "it was seen" by "it was revealed", and inserted "electromagnetic" after "coupled";
10) In Line 52, we added new reference [T.D. Kaladze, L.V. Tsamalashvili, L.Z. Kahlon, Rossby-Khantadze electromagnetic planetary vortical motions in the ionospheric E-layer, J. Plasma Phys., (2011), v. 77, 813-828]
11) At the end of line 68 we added the sentence on experimental evidence of planetary Rossby waves in the ionospheric E-region: "Some premises of the possibility of existence of planetary Rossby waves in the dynamo E-area of weakly ionized ionosphere and corresponding experimental interpretation was discussed by Forbes [Forbes, J.M., Planetary waves in the thermosphere-ionosphere system // J. Geomag. Geoelectr., v. 48, 91-98, (1996)]"
12) We removed the previous reference [19] and inserted new reference "19. Song Jian, Yang Lian-Gui, DA Chao-Jiu, and Zhang Hui-Qin, mKdV equation for the amplitude of solitary Rossby waves in stratified shear flows with a zonal shear flow, Atmospheric Oceanic Science Letters, 2009, #1, 18-23"
13) In the last paragraph of Introduction (lines 69-72) we made the following changes: "In previous publications [12, 16, 17, 18, new 19 by Song Jian], (1+1)D modified Korteweg-de Vries (mKdV) equations for the amplitude of solitary Rossby waves under the action of zonal shear flow derived in case of neutral fluids. In the given manuscript we generalized these results for the weakly ionized conducting ionospheric E-region plasma incorporating along with stream-function evolution of geomagnetic field for electromagnetic RK waves,

which to the best of our knowledge was not reported so far and thus provides novelty to this work." We think the motivation should be clear. Then continue : "In Sec. 2, by using …"

14) Answering your comment (V) we added: a) in line 193 the reference [16], b) instead lines 212-216 we changed: "Amid numerous exact solutions of mKdV equation (15) (see e.g. [Abdul-Majid Wazwaz, Partial Differential Equations and Solitary Waves Theory, Springer, 2009], we are interesting in soliton like traveling wave solution (next line is Eq. (18)), where c is the traveling wave velocity, and the coefficients D, and N are defined by Eqs. (16)-(17)".

15) Concerning the origin of Eq. (1) we added new reference [T.D. Kaladze, L.Z. Kahlon, L.V. Tsamalashvili, Excitation of zonal flow and magnetic field by Rossby-Khantadze electromagnetic planetary waves in the ionospheric E-layer, Phys. plasmas, v.19, 022902 (2012); doi: 10.1063/1.3681370] and instead lines 80-104 we changed the following:

We consider weakly ionized E-ionospheric region comprising of small concentration of electrons, ions and bulk of neutral particles, where the ionospheric plasma is enclosed in a spatially inhomogeneous geomagnetic field $\boldsymbol{B}_0 = (0, B_{0y}, B_{0z})$ and the Earth's angular velocity $\boldsymbol{\Omega} = (0, \Omega_{0y}, \Omega_{0z})$. In such E-layer therefore, two-dimensional consideration of the wave motion provides complete information about its propagation in terms of stream function $\psi(x, y, t)$, $\mathbf{v} = (u, v, 0)$, with $u = -\frac{\partial \psi}{\partial y}$ and $v = \frac{\partial \psi}{\partial x}$ .

We introduce a local Cartesian system of coordinates with zonal x, latitudinal y, and z in local vertical direction. Then the nonlinear behavior of the sheared electromagnetic Rossby-Khantadze waves can be described by the following system of 2D equations,

$$\begin{cases} \frac{\partial \Delta \psi}{\partial t} + \beta \frac{\partial \psi}{\partial x} + J(\psi, \Delta \psi) - \frac{1}{\mu_0 \rho} \beta_B \frac{\partial h}{\partial x} = 0 , \\ \frac{\partial h}{\partial t} + J(\psi, h) + \beta_B \frac{\partial \psi}{\partial x} + c_B \frac{\partial h}{\partial x} = 0 , \end{cases} \quad (1)$$

The first equation is the z-component of vorticity ($\zeta_z = \boldsymbol{e}_z \cdot \nabla \times \mathbf{v} = \Delta \psi$) of the single-fluid momentum equation under the action of the geomagnetic field, $\mathbf{v}$ is the velocity of the neutral incompressible gas. The second equation is the z-component of the perturbed magnetic induction h obtained through Faraday's law, and $\beta = \frac{\partial f}{\partial y} = \frac{2 \partial \Omega_{0z}}{\partial y}$ describes the latitudinal inhomogeneity present in the vertical component of angular velocity. Also the parameter $c_B = \beta_B / e n \mu_0$ with $\beta_B = \frac{\partial B_{0z}}{\partial y}$, describes the latitudinal inhomogeneity in the background magnetic field, $n$ is the number density of the charged particles, $\mu_0$ is the magnetic permeability and $J(a, b) = \frac{\partial a}{\partial x} \frac{\partial b}{\partial y} - \frac{\partial a}{\partial y} \frac{\partial b}{\partial x}$ is the Jacobian (responsible for the vector nonlinearity) and $\Delta = \partial_x^2 + \partial_y^2$ . Note that the small concentration of charged particles (compared to the neutral particles) gives the contribution only in the inductive current (Kaladze, et al. 2013). It should also be noted that the ambient magnetic field and Coriolis parameter are spatially inhomogeneous, $f = 2 \Omega_{0z}$ (Kaladze, et al., 2014). Details on the system (1) can be found in [T.D. Kaladze, L.Z. Kahlon, L.V. Tsamalashvili, Excitation of zonal flow and magnetic field by Rossby-Khantadze

electromagnetic planetary waves in the ionospheric E-layer, Phys. plasmas, v.19, 022902 (2012); doi: 10.1063/1.3681370]

16) As a private communication, we would like discuss your comment (IV). For the scientists engaged with the Rossby waves it will be clear that the first equation of the system (1) is written for the relatively small-scale structures $a \leq r_R$, where $r_R$ is the Rossby radius. With the opposite inequality the first term should be added by $-\frac{\partial}{\partial t} \frac{1}{r_R^2} \psi$.

Then obtained mKdV equation will contain quadratic nonlinear term (see Reference [18] by Shi et al.).

17) Remain comments on possible experimental observations are elucidated in Discussion, which should be read as follows:

**3. Discussion**

[revised manuscript text omitted]

**Yours Sincerely,**

**Laila Zafar Kahlon**

---

## Author Response (AR3)

Dear Editor,

According to comments of editor and refrees, we have made the experimental evidences, also references have been added specifically Kevrekidis et al, describing the properties of mKdV waves. Also, the numerical properties of RK waves make possible experimental observations. Throughout, the manuscript the English has been improved by foreigner.

.

Specifically, we have made the following changes in our manuscript according to your comments:

1) In line 19, we remove the following words "in the ionospheric in the ionospheric E-layer"
2) In line 21, we add the following words "with cubic nonlinearity".
3) In line 29-82, we rephrase the "Introduction"
4) In line 90-93, we changed the "In such E-layer therefore, two-dimensional consideration of the …….." with "In weakly ionized ionospheric E-layer plasma, we consider …" and "here" with "and".
5) In line 92, the word "with" is replaced with "where"
6) In line 98, we changed "… is the z-component of vorticity …" with "… describes the evolution of the …"
7) In line 102, we changed "… inhomogeneity present in the vertical component of angular velocity" with "latitudinal inhomogeneity of angular velocity";
8) In line 105, the following reference "Kaladze, et al. 2013" is replaced with "Kaladze, et al. 2013a, 2013b"
9) In lines 111-116, the repeated lines are removed.
10) In line 118, the word "in" is replaced by "for".
11) In line 122, we changed "[1, 2]" by "(Pedlosky (1987); Satoh (2004))."
12) In line 126, we changed "considered as" with "considered in the following manner:"
13) In line 131, we removed the following words "along with a normalized small parameter $\varepsilon \ll 1$."
14) In line 132, we changed the following words "Which forms" with "This assumption makes it".
15) In line 137, we add the following line "where $\varepsilon \ll 1$ is a small parameter indicating that the perturbed quantities are small compared to the background parameters." after Eq. (4).
16) In line 150, we changed the word "throughout" with "In" and following wording ", and the following linear differential operators are introduced" have been added at the end of sentence.
17) In line 157, the word "involves" is replaced by "takes into account".
18) In line 158, the word "inhomogeneous" is replaced by "inhomogenity of the"
19) In line 158-159, the following reference "Kaladze et al. [16]." is modified by Kaladze et al. (2009)."
20) In line 161, we rewrite "Furthermore, we denote the disturbed stream function $\psi$ (the asymptotic expansion) as: " as "Furthermore, we expand the stream function $\psi$ (in series with respect to the $\varepsilon$) as:"

21) In line 165-166, we rewrite "........., from the lowest order $O(\varepsilon)$, we get the following equation with conditions of boundary:" as "By using Eq. (8) into Eq. (6), we obtain from the lowest order $O(\varepsilon 0)$, the following equations,"

22) In line 191-192, the following line "Here it is pointed out that the dispersion effect competes with weakly nonlinear effect." Is replaced by "....., given in the definition of ............, which appears through the Jacobian in Eq. (9)."

23) In line 198, after "we" the following words have been added "again perform a separation of variables,"

24) In line 204-205, we change the following sentence "...results in the ... " with "we get the modified KdV (mKdV) equation"

25) In line 205, the following reference "[17]." is replaced by "(Kaladze et. al (2009))"

26) In line 208, after Eq. (15), the following sentence "This equation has a cubic nonlinearity, whereas the standard KdV equation has a quadratic nonlinearity."

27) In line 212, the word "where" has been added.

28) In line 215-216, the following lines "... equation (15) and ..." have been changed by "with cubic nonlinearity for Rossby waves ..."

29) In line 217, we have been changed "... point out here that the functions ..." with "...point out here the meridional dependence of functions..."

30) In line 217, we replaced the following word "are related" with "that appears in the"

31) In line 220, the reference "Wazwaz A-M., 2009" is replaced with "Wazwaz (2009))".

32) In line 221, the words "interesting in soliton" is replaced with "interested in a"

33) In line 221, the following line "The single soliton solution of equation (15) is given as ..." is added.

34) In lines 228, after "(16)-(17)." The followings lines "In order for a wave to have an exact solitary solution associated to it, one needs to a robust equation like the KdV. Modified KdV, as well, has infinite conservation laws associated to them, and hence is integrable ..."

35) In line 233, the word "given" is replaced with "present"

36) In line 233, the word "have" has been added.

37) In line 236, the word "latitudinal" have been added.

38) In line 239, the word "-" have been added twice.

39) In line 241, the word "modified KdV" is changed with "mKdV"

40) In line 246, the word "spatial inhomogeneous" is changed "the spatially inhomogeneous"

41) In line 248, the word "Looking forward" is changed with "In anticipation of future"

42) In line 249, we change the word "give" with "expect the"

43) In line 252, the word "the" has been added.

44) In line 252, the word "they" is replaced with "RK waves"

45) In line 254, we changed the words "charged particles' density" with "the number density of the charged particles"

46) In line 255, the following sentence "(which is so suitable for experimental observations)" has been added after "during the daytime and nighttime conditions"

47) In line 256, we changed the words "the wavelength" with "have wavelengths"

48) In line 265, we changed the words "Discussed" with "RK"

49) In line 268-269, we changed the words "will collect" with "provide"

50) In references [4-10, 12-13, 15-18, 22, 24-26, 28], the word "&" is throughout replaced with "and".

51) In line 294, we remove the reference 3.

52) In line 307, we remove the reference 8.

53) In line 314, we remove the reference 11.

54) In line 314, after reference 10 we add new reference "Futatani S., Horton W., and Kaladze T.D., Nonlinear propagation …"

55) In line 333, we shifted the reference [22] here, also two new references "Kahlon L.Z., and Kaladze T.D., Generation of zonal flow …" , also "Kaladze T., Tsamalashvili L., Kaladze D., Ozcan O., Yesil A., and Inc M., Modified …" has been added after it.

56) In line 335, we remove the reference [20] and add the following reference "Vukcevic M. and Popovic L. Č., Solitons in the ionosphere-Advantages and …"

57) In line 344, we add the following reference "Kevrekidis P.G., and Khare A., Saxena A., Herring G., On some classes of mKdV …"

**Yours Sincerely,**

**Authors**

---

## Author Response (AR4)

Dear Editor,

The format of references has been improved as per journal style.

**Yours Sincerely,**

**Authors**